# Identification of novel regulators of STAT3 activity

**Elina Parri**[1], **Heikki Kuusanmäki**[1,2], **Arjan J. van Adrichem**[1], **Meri Kaustio**[1],
**Krister Wennerberg**[1,2]*

**1** Institute for Molecular Medicine Finland, University of Helsinki, Helsinki, Finland, **2** Biotech Research & Innovation Centre (BRIC) and Novo Nordisk Foundation Center for Stem Cell Biology (DanStem), University of Copenhagen, Copenhagen, Denmark

* krister.wennerberg@bric.ku.dk

## Abstract

STAT3 mediates signalling downstream of cytokine and growth factor receptors where it acts as a transcription factor for its target genes, including oncogenes and cell survival regulating genes. STAT3 has been found to be persistently activated in many types of cancers, primarily through its tyrosine phosphorylation (Y705). Here, we show that constitutive STAT3 activation protects cells from cytotoxic drug responses of several drug classes. To find novel and potentially targetable STAT3 regulators we performed a kinase and phosphatase siRNA screen with cells expressing either a hyperactive STAT3 mutant or IL6-induced wild type STAT3. The screen identified cell division cycle 7-related protein kinase (CDC7), casein kinase 2, alpha 1 (CSNK2), discoidin domain-containing receptor 2 (DDR2), cyclin-dependent kinase 8 (CDK8), phosphatidylinositol 4-kinase 2-alpha (PI4KII), C-terminal Src kinase (CSK) and receptor-type tyrosine-protein phosphatase H (PTPRH) as potential STAT3 regulators. Using small molecule inhibitors targeting these proteins, we confirmed dose and time dependent inhibition of STAT3-mediated transcription, suggesting that inhibition of these kinases may provide strategies for dampening STAT3 activity in cancers.

## Introduction

STAT3 is constitutively activated in many aggressive cancers where it promotes drug resistance, an inflammatory phenotype and poor clinical outcome [1–3]. Aberrant STAT3 activation in cancer cells mainly occurs through canonical signalling when activated receptor- or non-receptor tyrosine kinases phosphorylate STAT3 at the tyrosine (Y) 705 residue. This results in dimerization of STAT proteins and localization to the nucleus where the STAT dimers bind to specific binding elements in the DNA and induce transcription of their target genes [3, 4]. In cancer, sustained STAT3 activation occurs through several mechanisms, such as cytokine stimulation, mutated cytokine receptors, mutated kinases (such as JAK family kinases), activation of non-receptor kinases (such as Src family kinases) or loss of negative regulators (such as SOCS and PIAS proteins) [5]. However, in rare cases such as large granular lymphocytic leukaemia, aggressive natural killer cell leukaemia and inflammatory

**Data Availability Statement:** All relevant data are within the paper and its Supporting Information files.

**Funding:** This project was supported by grants from: Academy of Finland (277293) (KW) https://

www.aka.fi/en Sigrid Jusélius Foundation (KW) https://sigridjuselius.fi/en/ Cancer Society of Finland (KW) https://www.cancersociety.fi/ In addition, this project was supported by University of Helsinki/Biocenter Finland Research infrastructure funds. The funders had no role in study design, data collection and analysis, decision to publish, or preparation of the manuscript.

**Competing interests:** The authors have declared that no competing interests exist.

hepatocellular adenocarcinoma, activating STAT3 mutations have been found in the Src homology 2 (SH2) domain near the Y705 phosphorylation site that result in a constitutively activated STAT3 [6–10]. From the mutations in the SH2 domain, STAT3(Y640F) causes increased hydrophobicity of the STAT dimerization site (pY+3 pocket) that results in more stable form of STAT3 dimer, constitutive Y705 phosphorylation, enhanced nuclear stability and increased transcriptional activity [6, 7, 9–11].

Since STAT3 is frequently hyperactivated and is thought to mediate malignancy, it is considered as potential therapeutic target for several solid and haematological cancers [12–14]. This has sparked multiple efforts to generate inhibitors, primarily aimed at blocking the SH2-phosphotyrosine dimerization interaction. As a protein-protein interaction, this is a challenging drug target but at least four small-molecule inhibitors have entered early-phase clinical testing [TTI-101 (NCT03195699), WP1066 (NCT01904123), OPB-51602 (NCT02058017, NCT01423903), OPB-31121 (NCT00955812)]. However, most (if not all) reported SH2 binding STAT3 inhibitors are electrophilic resulting in reactivity with cysteine thiols and they therefore have low selectivity towards the SH2 domain in more complex biological settings and often cause off-target toxicities [14–17]. An alternative approach is to inhibit the upstream signals leading to STAT3 activation, including more readily druggable targets such as kinases and cell surface receptors for which there are several approved and investigational agents [5]. Targeting the upstream signalling comes with limitations since it requires a specific aberrant upstream signalling cascade that is known. Furthermore, these approaches do not work well when there are activating mutations in the STAT3 protein itself [9, 18].

Here we set out to test whether inhibition of other, previously unexplored druggable targets can achieve inhibition of STAT3 signalling. First, we showed that constitutively active STAT3 protects cells from cytotoxic drug responses of several drug classes including anti-mitotic inhibitors and inhibitors of apoptosis proteins. Second, using reporter cells expressing either STAT3 wild type (wt) or constitutively active mutant STAT3(Y640F) in an RNAi screen, we identified kinases whose knockdown resulted in altered STAT3-activity. Finally, we confirmed that small molecule inhibition of several of these kinases could block STAT3 activity. Inhibition of these kinases may therefore be alternative strategies to inhibit STAT3 in cancers and other pathological conditions.

## Materials and methods

### Cell lines, virus production and transduction

HEK293 GloResponse SIE Luc2P Hygro cells (Promega) stably expressing a STAT3-dependent firefly luciferase reporter (here called HEK293sie cells) were grown in DMEM with 10% fetal bovine serum and 200 μg/ml Hygromycin B (Life Technologies). Cells were maintained at 37˚C with 5% $CO_2$ in a humidified incubator. HEK293sieSTAT3(Y640F) and HEK293sie-STAT3(wt) cells were generated as previously described [7]. Generation of lentivirus particles and cell transduction was performed as previously described [18]. All cell lines were grown to larger volume to make assay ready cells and frozen as aliquots in liquid nitrogen. After thawing the assay ready cells, they were maintained in culture conditions without antibiotics. All cell lines were tested negative for mycoplasma using the method described by Choppa *et al.* (performed by the THL Biobank, Helsinki, Finland) [19, 20].

### Drug sensitivity and resistance testing

Drug sensitivity and resistance testing (DSRT) was adapted to screen STAT3(wt) and STAT3 (Y640F) mutant overexpressing HEK293sie cells [21]. The screen consisted of 526 approved and investigational oncology compounds (S1 File). All compounds were pre-plated at 5

different concentrations in 10-fold dilutions to black wall clear bottom 384-well plates (Corning, cat. 3712) using an acoustic dispenser (Echo 550, Labcyte Inc.). 0.1% dimethyl sulfoxide (DMSO) and 100 μM benzethonium chloride were used as negative and positive controls, respectively. The list of tested compounds and calculated drug responses can be found in supplementary data 3. The pre-plated drugs were dissolved in 5 μl in cell culture medium with 1:400 CellTox Green (Promega) per well. Subsequently, 20 μl cell suspension (100,000 cells/ml) per well was dispensed into each well. Cells were incubated for 72 hours, at 37˚C and 5% $CO_2$. After 72 hours, cell death and viability were measured in multiplexed manner as described previously [22]. First, CellTox Green (Promega; cytotoxicity readout) was measured using fluorescence (485/520 nm excitation/emission filter) after which 25 μl CellTiter-Glo (Promega; viability readout) was dispensed per well, incubated for 10 minutes at room temperature and measured for luminescence. Both toxicity and viability were measured with a PHER-Astar FS plate reader (BMG Labtech). For the time point experiments, cells were plated in 20 μl/well (100,000 cells/ml) and drugs were added in 5 μl after 0, 48, 64 and 68 hours (for 72, 24, 8, 4-hour perturbation) from cell plating. HEK293sieSTAT3(wt) cells were induced with IL6 (50 ng/ml) (eBioscience) 3 hours before readout. 72 hours after cell plating, cell viability and reporter activity were measured with CellTiter-Fluor and ONE-Glo reagents (Promega) according to the manufacturer's protocol in multiplexed manner and read with a PHERAstar FS plate reader (BMG Labtech).

## siRNA screens

Chemically synthesized siRNAs from Ambion Silencer® Select Human siRNA Library V4 (Thermo Fisher Scientific) validated human kinase and phosphatase siRNA library (S2 File) consisting of 710 kinases and 298 phosphatases were dispensed into 384-well white clear bottom plates (Corning, cat. 3707) in 250 nl using an acoustic dispenser (Echo 550, Labcyte Inc.). Pre-dispensed plates were foil sealed and stored at -20˚C until use. In the first screen three siRNAs targeting one gene were pooled to one well. In all subsequent screens three individual siRNAs targeting one gene were in separate wells. Validation siRNAs were obtained from Qiagen (S3 File). All Ambion siRNA library and Qiagen siRNA screens were performed with the same protocol. Cells were reverse-transfected with library siRNA (10 nM, for pooled screen 9 nM) and Lipofectamine RNAiMAX Transfection Reagent (Thermo Fisher Scientific, cat. 13778150) using automated transfection as follows: Pre-plated siRNA plates were thawed to room temperature. siRNAs were complexed by addition of 5 μl 1:100 pre- diluted Lipofectamine RNAi-MAX in OptiMEM (Invitrogen) using a Multidrop Nano dispenser (Thermo Fisher Scientific) followed by incubation for 20 minutes at room temperature on a plate shaker (400 RPM). Trypsinized and re-suspended cells were plated in 20 μl yielding 2000 cells/well and 10 nM siRNA concentration. Assay plates were incubated at 37˚C and 5% $CO_2$, for 72 hours. Wild type STAT3 cells were induced with IL6 (50 ng/ml) (eBioscience) 3 hours before readout; 2 μl of diluted IL6 was dispensed using a Multidrop Nano. Subsequently, cell viability and reporter activity were measured with CellTiter-Fluor and ONE-Glo (Promega) according to the manufacturer's protocol in a multiplexed manner and read with a PHERAstar FS plate reader (BMG Labtech).

## IFNγ activation site (GAS) response element assays

One million HEK293 cells (Promega) were plated on a T25 dish. After 8 h, cells were transfected with 5 μg of a GAS reporter vector (pGL4[luc2P/GAS-RE/Hygro], Promega) and 15 μl FuGENE HD (Promega, cat no. E2311) according to the manufacturer's instructions. 16 hours post-transfection, cells were plated on pre-plated siRNAs as described above. Cells were

incubated on siRNAs for 48 hours before activation with 100 ng/ml IFNγ (Peprotech) after which incubation was continued for 24 hours. Subsequently, cell viability and reporter activity were measured with CellTiter-Fluor and ONE-Glo reagents (Promega), respectively.

## Western blotting

300,000 or 250,000 cells were pre-plated on 6-well plates and treated with drugs for 24, 48 or 72 hours. Compounds and their vendors are listed in supplementary file (S3 File). Cells were lysed in RIPA buffer (Cell Signaling Technology) with 1 mM PMSF. Lysates were briefly soni-cated (5 × 2 s) to shear the DNA, the amount of protein was determined using a Qubit™ Fluo-rometer (Life Technologies; Thermo Fisher Scientific), after which 25–50 μg of protein was loaded per well on 10% SDS-PAGE gels (Bio-Rad). Proteins were run on SDS-PAGE gel for 60 minutes at room temperature (200V) in 1x Tris/Glycine/SDS (cat. 1610772, BioRad) and transferred to methanol activated (1 minute at room temperature) PVDF membranes (Invitro-gen) over night at 4˚C (90 mA) in Towbin buffer (25 mM Tris, 192 mM glycine) with 20% methanol. The membranes were blocked in TBS-Tween (150 mM sodium chloride, 50 mM TRIS-HCl, pH 7.6 and 0.05% Tween 20) with 5% bovine serum albumin (BSA) for 1 hour at room temperature. Primary antibodies anti-STAT3 (cat. 2972, Cell Signaling Technology), anti-pSTAT3 Y705 (cat. ab76315, Abcam), anti-pSTAT3 S727 (cat. 9134, Cell Signaling Tech-nology) and anti-α-tubulin (cat. T9026, Sigma Aldrich) were diluted 1:1000 in TBS-Tween + 5% BSA. Anti-phospho epitope antibodies were probed overnight at 4˚C and total protein antibodies for 1 hour at room temperature on a rocking plate. Membranes were subsequently washed with TBS-Tween (3 × 5 min) and probed with LI-COR compatible secondary antibody at 1:15,000 dilution in TBS-Tween (Odyssey IRdye 800CW and IRdye680) for 1 hour at room temperature. Membranes were visualized with an Odyssey imaging system (LI-COR Biosci-ences) and ImageJ version 1.47v was used to quantify band intensities.

## Data analysis

DSS, a quantitative drug sensitivity score, which is based on the area under the dose response curve, was used to measure drug efficacies [21, 23]. The DSS values were calculated in the Breeze web-based application [24].

Reporter activity of siRNA in primary screens were normalized to cells (negative control, 100%) and siSTAT3 (positive control, 0%) and in validation assay to non-targeting siRNA (negative control, 100%) and siSTAT3 (positive control, 0%). Viability readouts were normal-ized to non-targeting siRNA and siDEATH.

Statistical analyses were performed with Prism software version 8.0 (GraphPad Software). Differences between responses were calculated by using the Student's t-test (unpaired t-test). P-values of <0.05 were considered as statistically significant.

## Results

### STAT3 activation protects cells from cytotoxic drug responses

To test whether STAT3 hyperactivity causes resistance to certain drug classes we used HEK293 cells expressing a firefly luciferase under a STAT3 responsive sis-inducible element (SIE) pro-moter (HEK293sie cells) and either STAT3(Y640F) mutant or wild type STAT3. Constitutively active mutant STAT3(Y640F) allowed us to selectively measure the effect of drug responses in a state of high STAT3 activation. We tested 526 oncology compounds in drug sensitivity and resistance profiling (DSRT) and measured cell toxicity by a cell-impermeable DNA-binding dye and viability by a cellular ATP detecting reagent in a multiplexed manner after a 72-hour

treatment (S1 File). Subsequently, we calculated a drug sensitivity score (DSS) that measures the area under the dose response curve and represents the efficacy and potency for each compound [21, 23]. When we compared the DSS values of STAT3(wt) and STAT3(Y640F) expressing cells we did not see clear differences in the viability readout (Fig 1A). In the cell toxicity readout, on the other hand, we observed that STAT3(Y640F) is protecting cells from cytotoxic responses against several drug classes (Fig 1B), such as a selective inhibitor of exportin-1 (XPO1), microtubule targeting agents, Polo like kinase 1 (PLK1), heat shock protein 90 (HSP90) and cyclin-dependent kinase (CDK) inhibitors (Fig 1C and 1D). This suggests that

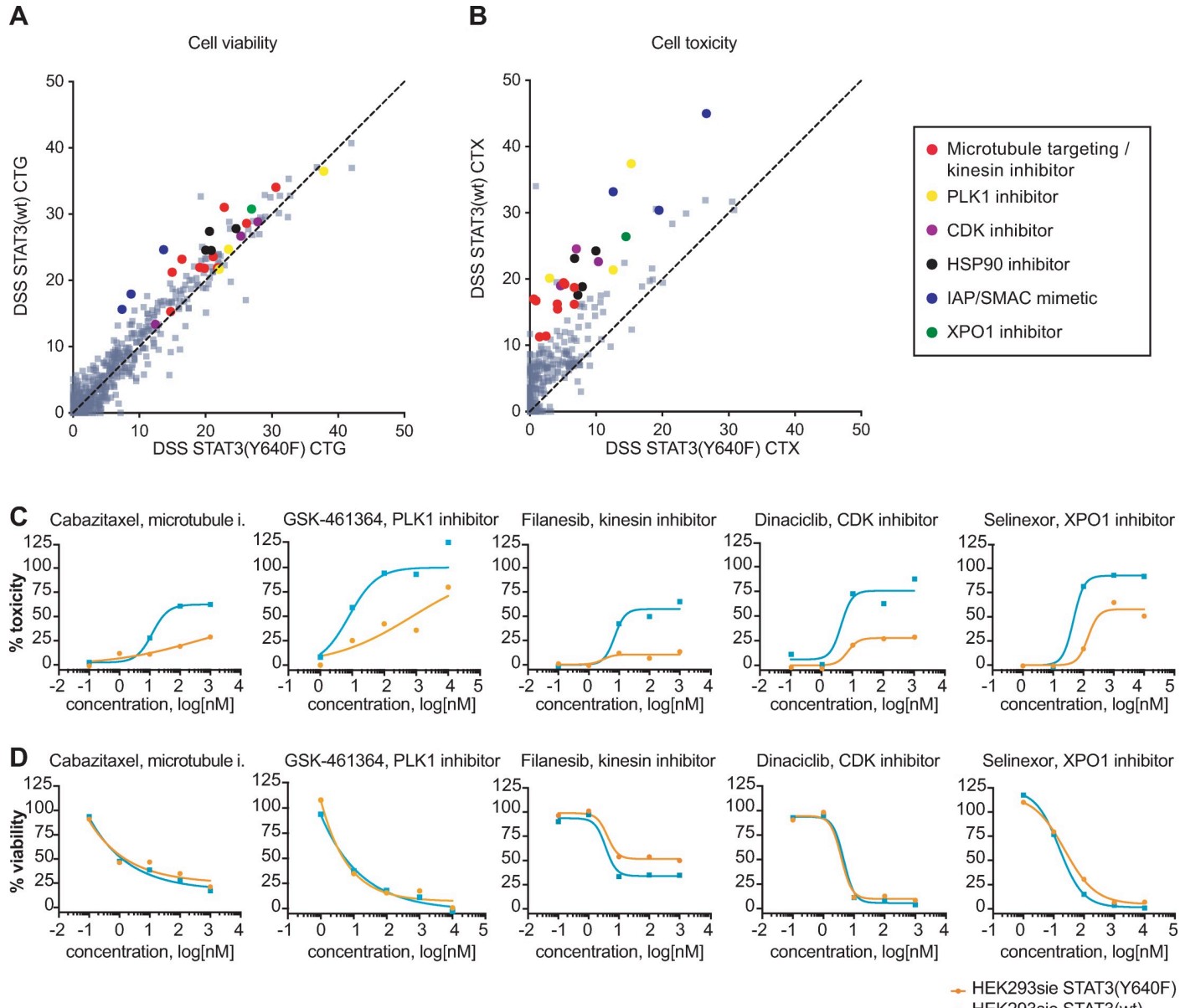

**Fig 1. Constitutive activation of STAT3 protects cells from several drug classes.** (A) In an ATP-based cell viability readout (CellTiter-Glo, CTG) chemoprotection is not perceptible when constitutively active STAT3(Y640F) was compared to STAT3(wt) expressing cells. (B) In a cytotoxicity readout, the constitutively active STAT3(Y640F) protects cells from cytotoxic effects of several drug classes. (C) Dose-response curves of percent toxicity (CellTox Green, CTX) and percent viability (D) of HEK293sie cells expressing STAT3(Y640F) or STAT3(wt). Data from one representative experiment out of two independent experiments shown.

the constitutively active STAT3 protected cells from the cytotoxic, but not the cytostatic effects of these drug classes.

## siRNA screening reveals kinases and a phosphatase that regulate oncogenic hyperactive STAT3

To test how hyperactive STAT3 is regulated we designed an siRNA screen using HEK293sie cells stably co-expressing STAT3(Y640F) or STAT3(wt). The STAT3(Y640F)-expressing cells exhibited a constitutive STAT3 reporter activity while the activity in the STAT3(wt)-expressing cells could be induced by IL6 stimulation. The STAT3(Y640F) expressing HEK293sie cells had a stronger reporter signal than IL6-induced STAT3(wt) expressing cells (3.6-fold) (S1 Fig). Using these cells, we performed an siRNA screen where the impact of knockdown of pooled siRNAs (3 siRNAs per gene) targeting protein/lipid kinase and protein phosphatase genes on STAT3 transcriptional activity was assessed (Fig 2A, S2 File). STAT3 silencing, which almost totally abrogated the STAT3-mediated luciferase signal, was used as a positive control (S1 Fig). In addition, we evaluated the role of *JAK1* knockdown in the STAT3(wt) vs. STAT3(Y640F) expressing cells. While *JAK1* knockdown nearly completely blocked IL6-stimulated STAT3 (wt) transcriptional activity, it had only a partial effect on the STAT3(Y640F) signalling (Fig 2B), as has been reported before [9, 18]. From the primary screen consisting of 1,056 genes, we selected 182 genes, which were retested by assessing the effect of the three individual siRNAs in separate wells (Fig 2A, S2 File).

From these, we identified 25 hit candidate genes where at least 2 of the 3 individual siRNAs confirmed. Seven genes validated in a follow-up screen using three additional siRNAs from a different vendor (Fig 3A and 3B and S2 File). Knockdown of the kinase genes *CSNK2*, *CDC7*, *DDR2*, *CDK8*, *PI4KII* and *PTPRH* resulted in inhibition of both STAT3(Y640F) and STAT3 (wt) reporter signals (Fig 3A and 3B). Conversely, knockdown of *CSK* caused a significant

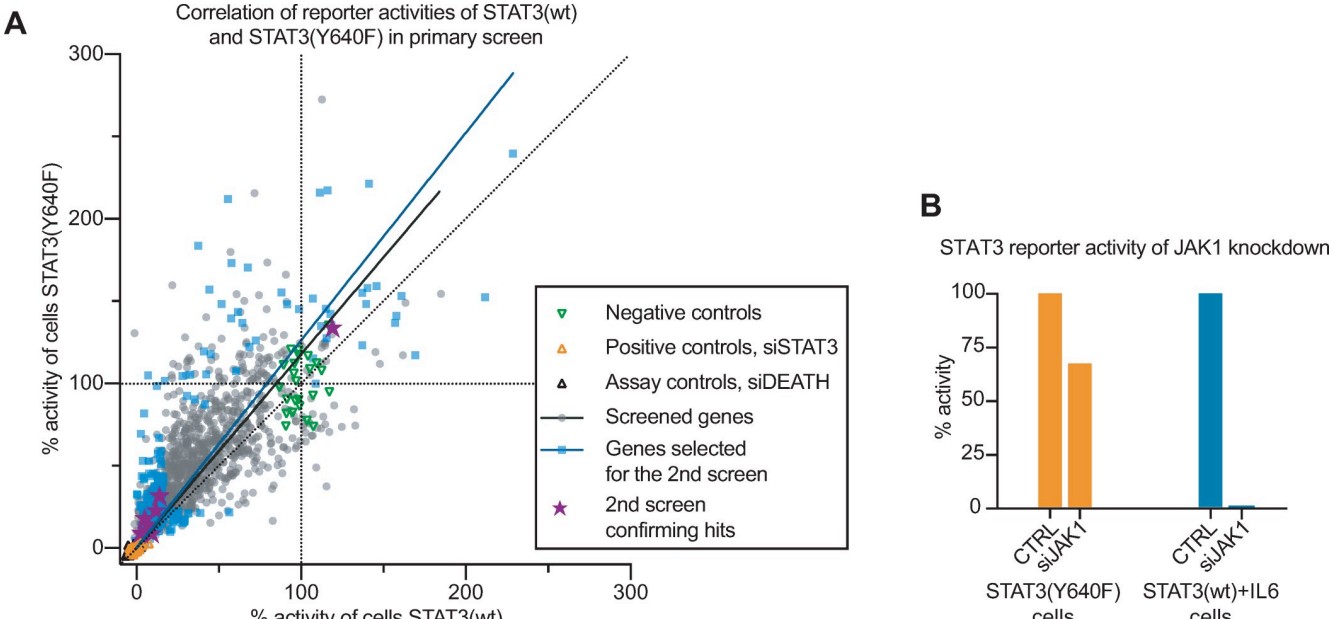

**Fig 2. Small interfering RNA (siRNA) screen to identify regulators of hyperactive STAT3.** (A) General distribution of the first screen with final validated hits is highlighted. From each screen, siRNAs reducing cell viability more than 2 times the standard deviation of the negative controls were excluded. In the validation screen, the siRNAs were obtained from separate source. (B) *JAK1* knockdown had a strong effect on STAT3(wt) transcriptional activity, but not in STAT3(Y640F) expressing cells. Reporter activity was normalized to positive (0%) and negative controls (100%), and cell viability (CellTiter-Fluor).

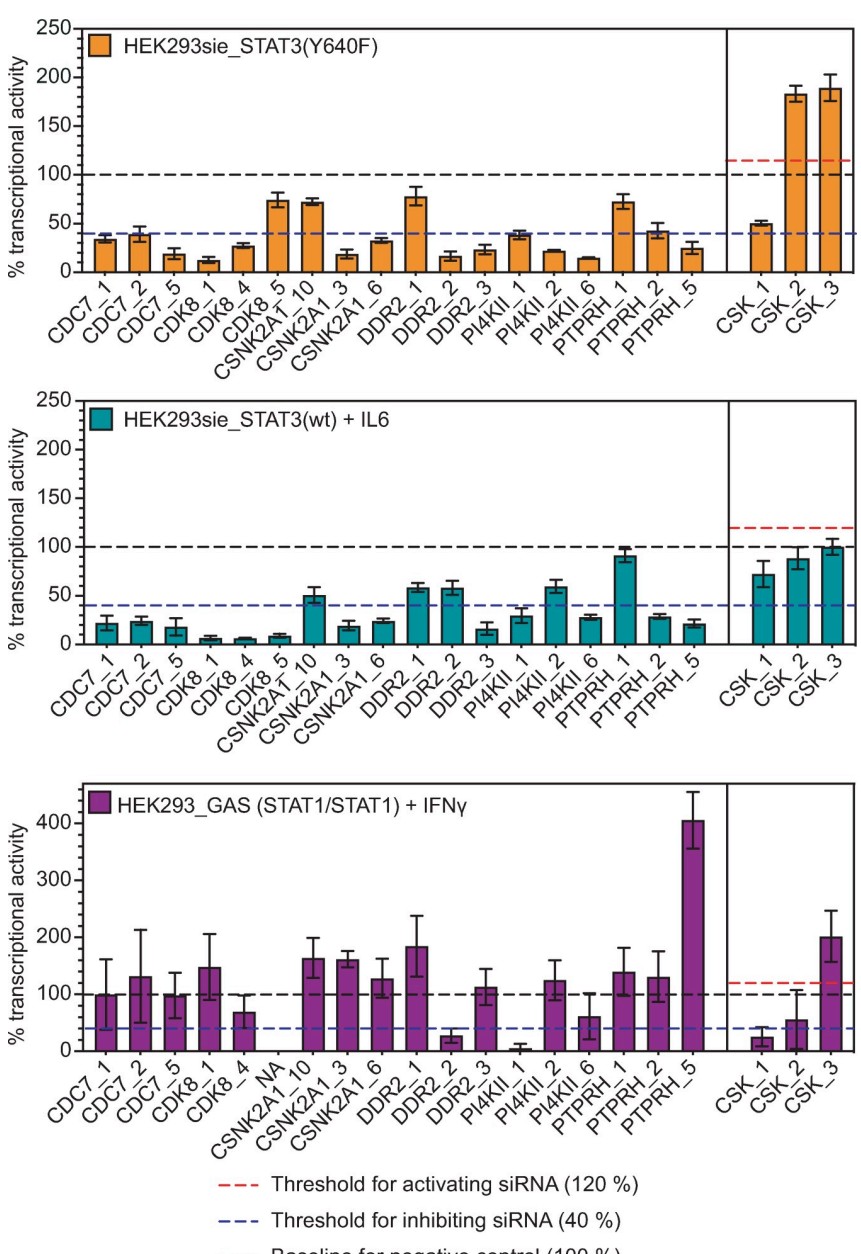

**Fig 3. Genes regulating STAT3(wt) or STAT3(Y640F) reporter activity.** (A-B) Hit genes whose knock-down resulted in changed transcriptional activity of either STAT3(Y640F) (A) or STAT3(WT) (B). Gene hits were normalized to siSTAT3 (positive control, 0% transcriptional activity) and non-targeting siRNA (negative control, 100% transcriptional activity). Red and blue lines represent thresholds (2 times standard deviation of control) for activating and inhibiting hits, respectively. Data comes from one representative experiment out of two independent experiments with three technical replicates. Bars represent mean ± SD. (C) Six out of seven hits were selective for STAT3. Only *PI4KII* knockdown inhibited both STAT3:STAT3 and STAT1:STAT1 transcriptional activity. Data comes from three independent experiments in triplicates. Two or three siRNAs were used against each gene.

increase of reporter activity in STAT3(Y640F), but not in the IL6-induced STAT3(wt) expressing cells, indicating that CSK could be a selective positive regulator of constitutively activated STAT3 signalling.

To determine whether the gene products of the seven hit genes were selectively regulating STAT3 signalling, we generated HEK293 cells that transiently expressed a luciferase reporter for STAT1-driven IFNγ activation site response element (GAS-RE) (Fig 3C). Only knockdown of *PI4KII* inhibited both STAT3 and STAT1 transcriptional activity. The rest of the knockdowns did either not have a significant effect on IFNγ activated STAT1 (*CDC7*, *CSNK2A1*, *DDR2*, *PTPRH*) or were non-conclusive (*CDK8*, *CSK*).

## Small molecule inhibitors of hit kinases regulate STAT3 activity in a dose and time dependent fashion

To further validate the hit kinases, we tested the effects of STAT3 reporter activity by small molecule inhibitors targeting the kinase activity of the hit proteins. We used ruxolitinib, silmitasertib, senexin B, BMS863233 and regorafenib to target JAK1/2, CSNK2, CDK8, CDC7 and DDR2, respectively. All inhibitors induced reduction of STAT3 reporter activity at earlier time points and lower concentrations than those the cause cytotoxic/cytostatic effects (Fig 4A–4J).

Using these compounds, we also assessed the temporal effects of kinase inhibition. We observed that the reporter activity inhibition occurred more slowly in STAT3(Y640F) expressing cells than the STAT3(wt) cells, suggesting that the STAT3(Y640F) protein de-phosphorylation is slower. However, this could also be due to the difference in assay settings: in the STAT3(wt) cells we blocked *de novo* activation of STAT3(wt) while in the STAT3(Y640F) case the inhibitors blocked sustained activation. In agreement with the siRNA data, the JAK1/2 inhibitor ruxolitinib did not inhibit mutant STAT3(Y640F) at any time point, whereas inhibition of IL6-induced STAT3(wt) was seen already after 4 hours (Fig 4A and 4F). The STAT3 reporter inhibition by the CSNK2 inhibitor silmitasertib was notably slower in the STAT3(Y640F) than in the STAT3(wt) cells, where a solid reporter inhibition was seen with the STAT3(wt) already after 4 hours and the corresponding response in the STAT3(Y640F) cells was only seen after 24 hours (Fig 4B and 4G). The CDK8 inhibitor senexin B showed slower reduction of reporter activity in STAT3(Y640F) than in IL6 induced STAT3(wt) expressing cells (Fig 4C and 4H). A similar result was seen with another CDK8 inhibitor; senexin A (S2 Fig). The CDC7 inhibitor BMS863233 led to a slow inhibition of STAT3 activity in both cell lines possibly indicating regulation at the transcriptional level. Due to the lack of a selective DDR2 inhibitor, we used the multikinase inhibitor regorafenib, which is known to be a potent DDR2 inhibitor [25]. Regorafenib has also been previously proposed to inhibit STAT3 through activation of SHP1 [26]. However, SHP1 (*PTPN6*) is primarily expressed in hematopoietic cells and we did not detect significant expression from RNA sequencing of our cells. Regorafenib was only the compound that clearly showed stronger potency towards STAT3(Y640F) expressing cells at 72 hours. Similar to silmitasertib, senexin B and BMS863233, regorafenib treatment resulted in faster decrease of STAT3 reporter activity in IL6-induced STAT3(wt) expressing cells than in STAT3(Y640F) cells.

We did not have potent or selective inhibitors against PTPRH, PI4KII and CSK and hence we could not validate these hits with small molecules. Based on the results from the primary siRNA screen on STAT3(Y640F) expressing cells, (S3 Fig) we hypothesized that CSK regulates STAT3 through inhibition of Src family kinases (SFKs) that in turn can phosphorylate and thus activate STAT3. SFK inhibition could therefore reduce transcriptional activity of STAT3(wt) and STAT3(Y640F) (S4 Fig). In agreement with this hypothesis, we observed that SFK inhibitors (saracatinib, CCT196969, dasatinib and A419259) resulted in stronger reporter activity reduction after 72-hour treatment in STAT3(Y640F) than in wild type STAT3 expressing cells, but also affected cell viability in both cell models (S4 Fig). The effect of SFK inhibition on cell viability was also observed in the siRNA screens and hence, the SFKs had been excluded

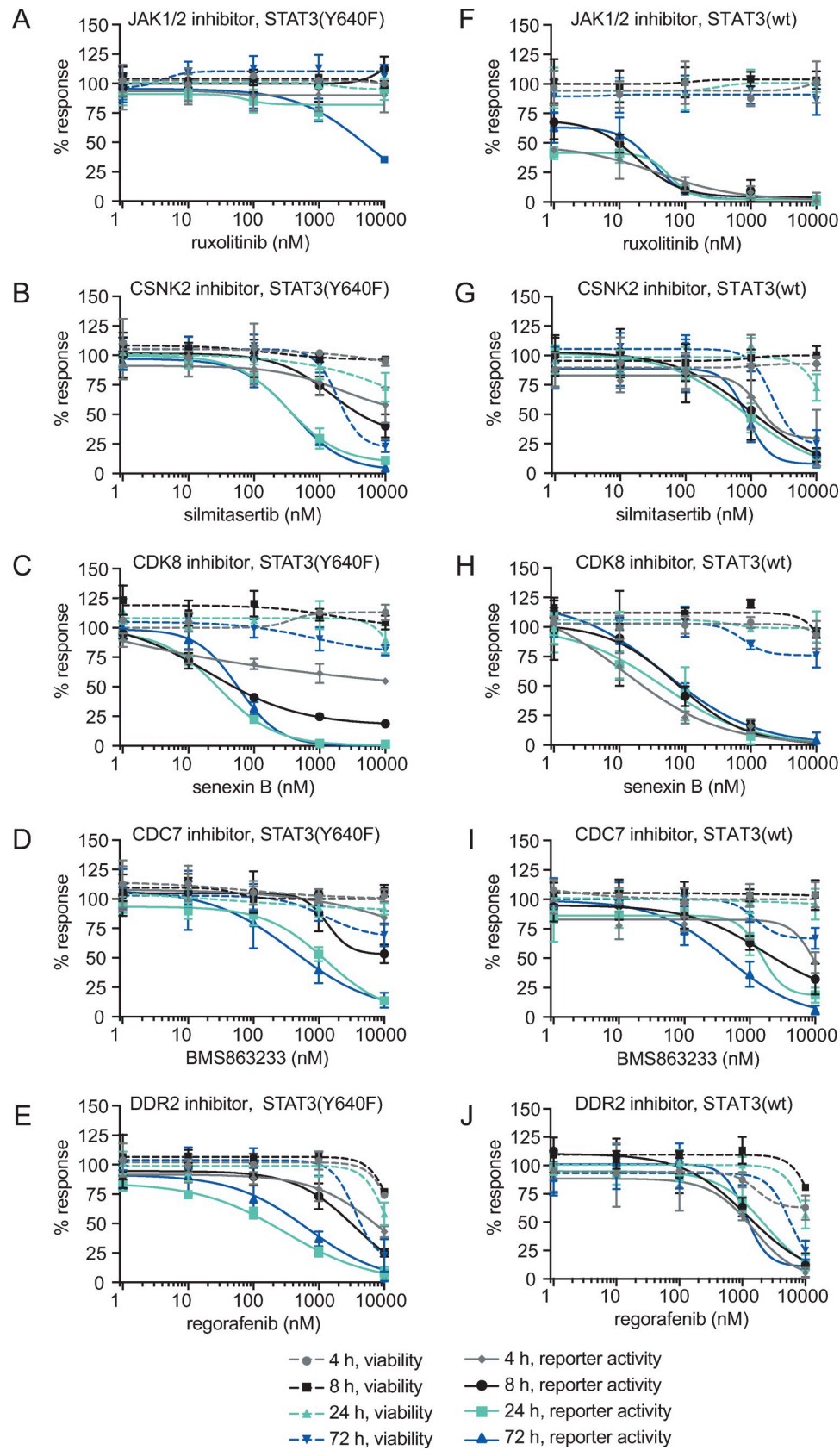

**Fig 4. Dose and time dependency of small molecule inhibitors to STAT3 expression.** Kinase inhibitor-mediated inhibition of STAT3 transcriptional activity is slower in cells expressing STAT3(Y640F) (A-E) than in cells expressing STAT3(wt) (F-J). STAT3(wt) cells were induced with 50 ng/ml IL6 3 hours prior to the reporter activity readout. Viability was measured in a multiplexed manner using CellTiter-Fluor. Error bars represent data from at least two independent experiments with two technical replicates.

as hits at various stages of the siRNA screening and validation path (Fig 2B, S3 Fig). We observed that knockdowns of *SRC* and *LCK* caused inhibition of transcriptional activity in STAT3(Y640F) expressing cells, but from these only *SRC* was highly expressed in HEK293sie-STAT3(Y640F) cells, suggesting that in our model, CSK may regulate STAT3(Y640F) through inhibition of SRC (S3 Fig).

Next, we investigated the mechanism of action of the selected kinase inhibitors and tested the effect of the drugs on STAT3 activity by measuring levels of phosphorylated Y705 STAT3 after compound treatment (Fig 5). We observed that senexin A, senexin B, regorafenib and BMS863233 caused reduction of pY705 in STAT3(Y640F) cells, but none of the compounds were able to completely abolish pY705. As expected, JAK1/2 inhibition with 1 µM ruxolitinib did not markedly reduce pY705 STAT3 level in STAT3(Y640F) expressing cells after 48 or 72 hours (Fig 5), but had a strong inhibitory effect on STAT3(wt) Y705 phosphorylation already after 4 hours (S5 Fig). We noted that even with as high a concentration as 1 µM, 48 to 72-hour treatment with the CSNK2 inhibitor silmitasertib did not cause as strong a reduction in pY705 levels of STAT3(Y640F) cells as previously published [27] (Fig 5). However, silmitasertib reduced S727 phosphorylation in STAT3(Y640F) expressing cells after 48-hour perturbations (Fig 5). The inhibition of CDK8 and CDC7 caused the strongest reduction of both pY705- and pS727-STAT3 in STAT3(Y640F) expressing cells whereas in IL6-induced STAT3(wt) expressing cells, only the JAK1/2 inhibitor reduced pY705-STAT3 and none of the inhibitors had an effect on pS727-STAT3. In addition, CDK8 inhibition affected the levels of total STAT3 protein both in STAT3(wt) and STAT3(Y640F) cells (Fig 5). The hit kinases and the inhibitors used are summarized in Fig 6.

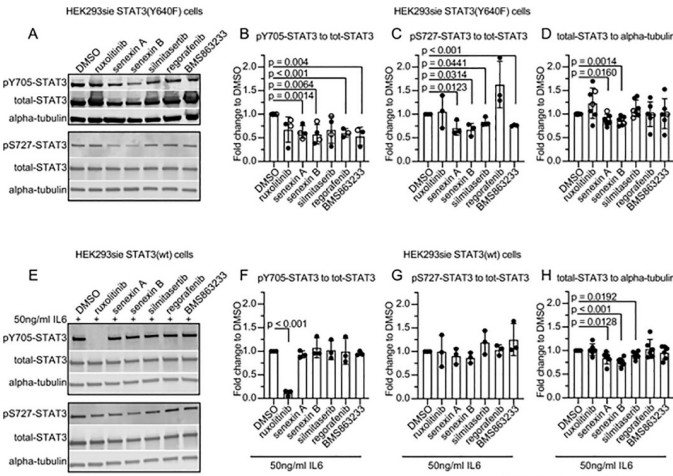

**Fig 5. The effect of kinase inhibitors on STAT3 levels in HEK293sieSTAT3(Y640F) or HEK293sieSTAT3(wt) cells.** pY705-, pS727- and total-STAT3 levels in STAT3(Y640F) expressing (A-D) and IL6-induced STAT3(wt) (E-H) cells after 48-72-hour perturbations with the kinase inhibitors. pY705 and pS747-STAT3 are blotted on individual membranes and normalized to total-STAT3 from the same membrane. their All the compounds were used at 1 µM concentration. Mean ± SD. Each dot represents data from an independent experiment. P-values were calculated using Student's t-test (unpaired).

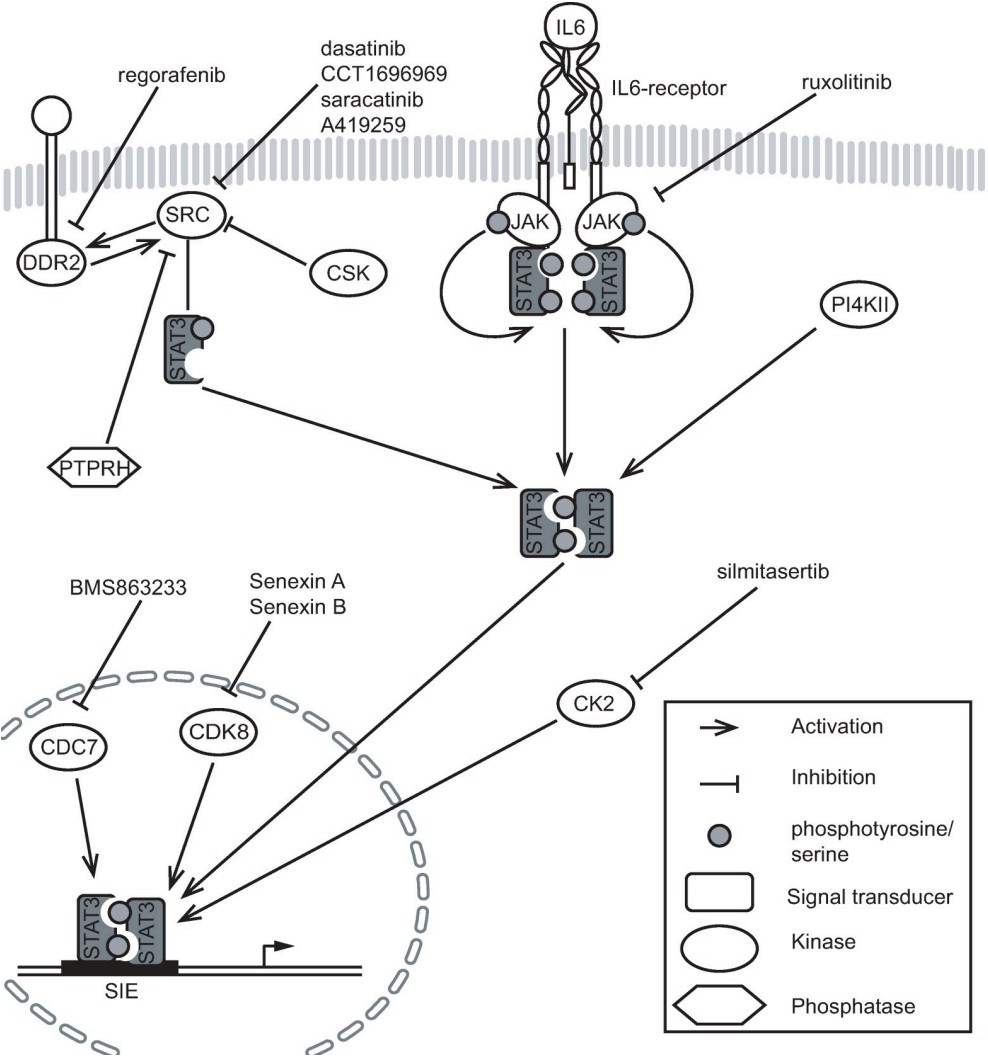

**Fig 6. Schematic illustration of hit kinases/phosphatase and inhibitors targeting the kinases.**

## Discussion

In this study, we sought to identify new druggable approaches to target STAT3 hyperactivity in cancers. First, to study whether hyperactive STAT3 influences therapeutic responses, we performed drug screening in cells expressing a constitutively active mutant STAT3. We discovered that STAT3 hyperactivity protects cells from undergoing cell death and observed STAT3-mediated cytoprotection towards multiple antimitotic as well as pro-apoptotic drugs. This was seen as an effect of reducing apoptotic, but not cytostatic effects. The lack of difference in cytostatic effects in STAT3(wt) and STAT3(Y640F) cells is in agreement with our previous report showing that in a strong viability response (with relatively fast-growing cells), it is often difficult to separate cytotoxic from cytostatic responses [22]. The observed cytostatic effects are in agreement with previous studies showing that STAT3 can protect cancer cells from the apoptotic effects of many individual drugs: STAT3 activity controls the expression of pro-survival genes such as *MCL1*, *BCL2* and *BCL2L1* [28, 29]. STAT3 activity can also upregulate members of the IAP protein family proteins such as survivin that prevents caspase activity

[30] and extrinsically supress FAS ligand mediated apoptosis [31]. Furthermore, STAT3 can play a role in regulating multidrug resistance gene expression [32].

We observed STAT3-mediated cytoprotection to several microtubule targeting agents. STAT3 activity has previously been associated with paclitaxel resistance in ovarian and colorectal cancer cell lines [33, 34]. Docetaxel resistance have been linked to constitutive STAT3 activation and elevated survivin expression in breast cancer [35]. STAT3 and PLK1 are known to control each other's transcription in a positive feedback loop resulting in cancer cell survival, proliferation and resistance to apoptosis [36]. Together this supports the concept that constitutive STAT3 activity is protecting cancer cells from apoptotic effects of mitotic inhibition. Cells expressing constitutively active STAT3 were also protected from cytotoxic effects of the exportin-1 inhibitor selinexor. exportin-1 inhibition with the selinexor analogue KPT-276 has been shown to repress STAT3 activation and survivin transcription in breast cancer cell lines [37].

Under normal conditions STAT3 signalling can be regulated by several tyrosine kinases and their upstream/extrinsic regulators (i.e. IL6, IL10, LIF, EGF). Widely known STAT3 activators include JAK family proteins, receptor tyrosine kinases such as EGFR and MET and non-receptor tyrosine kinases such as SFKs and Abl [38]. Negative regulators include SOCS that inhibit STAT3 phosphorylation by blocking the enzymatic activity of JAK or by competing with STAT3 of binding to the IL6 receptor complex [39, 40], protein tyrosine phosphatases that dephosphorylate STAT3 (such as PTPRT, PTPRD, SHP2, SHP1 and PTPN9), which dephosphorylate STAT3 [41], and PIAS proteins that block the DNA binding of STAT3 [42].

We found six kinases (CSK, CSNK2, CDC7, DDR2, CDK8, PI4KII) and a phosphatase (PTPRH) that indirectly regulate either oncogenic hyperactive STAT3(Y640F) mutant and/or IL6-induced wild type STAT3 in HEK293sie cells. From these, CSNK2 [27, 43], CDK8 [44] and CSK [45, 46] have been previously linked to STAT3 signalling. CDC7, DDR2 (also CD167b), PI4KII and PTPRH, on other hand, have not previously been associated with STAT activity. The identified kinase/phosphatase regulators were similar between STAT3(Y640F) and IL6-induced STAT3(wt), with the exception of CSK knockdown that resulted in an increased activation of STAT3(Y640F) mutant, but did not have a significant effect on IL6-induced STAT3(wt).

SFKs are known to regulate STAT3 phosphorylation independently of JAKs [47–49] and SFKs are negatively regulated by CSK by phosphorylation of a tyrosine residue at the C-terminal tail site [50]. Our data showed that knockdown of CSK resulted in upregulation of STAT3(Y640F) transcriptional activity, but did not have a clear effect on IL6-induced wild type STAT3 activity. This suggests that CSK may selectively regulate active STAT3(Y640F), but not IL6-induced wild type STAT3. In our cell models, inhibition of SFKs with small-molecule inhibitors reduced STAT3 reporter activity more efficiently in STAT3(Y640F) than in STAT3 (wt) expressing cells, but these inhibitors also had a strong effect on cell viability. In agreement with our results, it has been shown that SFK inhibition decreases STAT3(Y640F) activity in Hep3B cells [9].

*CSNK2* knockdown resulted in inhibition of both STAT3(Y640F) and IL6-induced wild type STAT3 reporter activity in our experiments. The CSNK2 inhibitor 4,5,6,7-tetrabromo-benzotriazole (TBB) has previously been shown to inhibit both IL6 and Hyper-IL6 induced STAT3 phosphorylation in Ba/F3 cells [27]. Furthermore, CSNK2 activity was shown to be required for STAT3(Y640F) activity [27]. However, in our assays CSNK2 inhibition with silmitasertib did not result to strong reduction of pY705 or pS727-STAT3 in either STAT3(Y640F) or IL6-induced STAT3(wt) cells suggesting that this could be cell line dependent.

CDK8 is a nuclear kinase that is directly involved in transcriptional regulation and is part of the Mediator complex [51]. We showed that CDK8 knockdown reduces transcriptional

activity of both hyperactive mutant and IL6-induced wild type STAT3. Nuclear CDK8 has been shown to regulate STAT3 and STAT1 S727 phosphorylation. It has been suggested that nuclear CDK8 activity and S727 phosphorylation increases the transcriptional activity of both STAT3 and STAT1 [44, 52–54]. Bancerek and coworkers showed that *CDK8* knockdown results in reduced levels of STAT3 S727 phosphorylation when induced with IFNβ, but they did not see changes in STAT3 Y705 phosphorylation [44]. We observed that CDK8 inhibition with 1 μM senexin A or B results in reduced Y705- and S727-STAT3 phosphorylation in STAT3(Y640F) cells, but not in IL6-induced STAT3(wt) cells. Furthermore, senexin A and B reduced total-STAT3 protein in both cell lines.

Our siRNA screen also identified three kinases and one phosphatase (CDC7, PI4KII, DDR2 and PTPRH) that have not previously been linked directly to STAT3. However, another discoidin homology domain-consisting RTK, DDR1, which is closely related to DDR2 has been linked to the regulation of STAT5 [55]. While PTPRH (SAP-1) has not been associated with STAT activity before, several other tyrosine phosphatases such as PTPN11 (SHP2), PTPRD, PTPRT (PTPrho) and PTPN6 (SHP1) have previously been found to act as negative regulators of STAT3 by dephosphorylating active STAT3 [41]. Our results on PTPRH suggest the opposite function, perhaps through removal of STAT3-inhibitory phosphorylations. PTPRH has been shown to be overexpressed in different adenocarcinomas, including colon, pancreas and non-small cell lung cancer [56–58].

In conclusion, our data shows that STAT3 hyperactivity protects cells from cytotoxic effects by several mitotic and pro-apoptotic drugs, arguing that STAT3 inhibition could be used to enhance the effects of standard chemotherapy. In addition, our data suggests that indirect targeting of STAT3 may be a feasible way to reduce constitutively active STAT3 in cancers instead of direct STAT3 inhibition. By using cells expressing the constitutively active STAT3(Y640F) mutant as a tool to study hyperactive STAT3, we show that small molecule inhibitors targeting CSNK2, CDK8, DDR2 and CDC7 may be used to target hyperactive STAT3 in cancer cells and warrant further validation of these targets in cancer models with STAT3 hyperactivity.

## Supporting information

**S1 Fig. Luciferase activities of controls in primary siRNA screen.** STAT3(Y640F) and IL6 induced STAT3(wt) expressing cells had 2.6-fold change in mean values of non-targeting siRNA (siASN) and 3.6-fold change in mean values of only transfection reagent (cells) treated cells. Bars represent mean and standard deviation of controls on all assay plates in one screen (total 24 technical replicates per each control). Cells and siSTAT3.7 were used to normalize the primary screen.
(TIF)

**S2 Fig. CDK8 inhibition in STAT3 expressing cells.** CDK8 inhibitor senexin A inhibits both STAT3(Y640F) and STAT3(wt) dose and time dependently. Dots are mean and error bars represent data from two independent experiments with two technical replicates.
(TIF)

**S3 Fig. Src family kinases in STAT3 expressing cells.** Effect of CSK and Src family kinases (SFKs) knockdowns on STAT3 reporter activity and viability of HEK293sieSTAT3(Y640F) (A) and IL6 induced HEK293sieSTAT3(wt) (B).
(TIF)

**S4 Fig. Effect of Src inhibitors on reporter activity of IL6 induced STAT3(wt) and STAT3 (Y640F) expressing cells.** Src inhibitors (saracatinib, CCT196969, dasatinib and A419259) results in stronger reporter activity reduction after 72h in STAT3(Y640F) that in wild type

STAT3 expressing. Dots are mean and error bars represent data from at least two independent experiments with two technical replicates.
(TIF)

**S5 Fig. Short term perturbation on Y705-phosporylation in IL6 induced STAT3(wt) and STAT3(Y640F) expressing cells.** Four-hour perturbation with JAK1/2 inhibitor ruxolitinib decreases Y705-phosphorylation of STAT3 in IL6 induced STAT3(wt) expressing cells.
(TIF)

**S1 File. Supporting file for Fig 1.** Drug sensitivity profiling data for viability (CTG) and toxicity (CTX) readouts, including in house ID, drug name, analysis name, DSS, IC50, drug concentration range (min/max conc.), percent inhibition value at each tested concentration (D1-D5), graph etc. of HEK293sieSTAT3(Y640) and HEK293sieSTAT3(wt) cells.
(XLSX)

**S2 File. Supporting file for Figs 2 and 3.** Normalized siRNA screening data for each screen including RefSeq accession number, gene ID, siRNA ID, full gene name gene symbol, percentage reporter activity normalized to positive (0%) and negative (100%) control and percentage cell viability normalized to positive (0%) and negative (100%) control. Each screen is on separate sheet.
(XLSX)

**S3 File. Supporting file for materials and methods.** Vendor information of siRNAs and siRNA screening controls used in Fig 3 and small molecule inhibitors used in Figs 4 and 5 and S1, S3, S4 Figs.
(XLSX)

**S1 Raw images. Raw Western blot scans.** Whole membranes of all shown and analyzed Western blot membranes with lane labelling. Lanes with "X" and/or "*italics*" are not included in the paper. Blots are supporting Fig 5 and S4 Fig.
(PDF)

## Acknowledgments

We gratefully thank Laura Turunen, Jani Saarela, Carina von Schantz-Fant and other members of the FIMM High Throughput Biomedicine unit for technical assistance with regards to screening.

## Author Contributions

**Conceptualization:** Elina Parri, Heikki Kuusanmäki, Arjan J. van Adrichem, Krister Wennerberg.

**Data curation:** Elina Parri, Krister Wennerberg.

**Formal analysis:** Elina Parri, Krister Wennerberg.

**Funding acquisition:** Krister Wennerberg.

**Investigation:** Elina Parri.

**Methodology:** Elina Parri, Heikki Kuusanmäki, Arjan J. van Adrichem, Meri Kaustio, Krister Wennerberg.

**Project administration:** Elina Parri, Krister Wennerberg.

**Resources:** Krister Wennerberg.

**Supervision:** Krister Wennerberg.

**Validation:** Elina Parri.

**Visualization:** Elina Parri.

**Writing – original draft:** Elina Parri, Krister Wennerberg.

**Writing – review & editing:** Elina Parri, Heikki Kuusanmäki, Arjan J. van Adrichem, Meri Kaustio, Krister Wennerberg.

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
