## [Decision Letter · Decision Letter 0]

2 Jan 2020

PONE-D-19-33342

Identification of novel regulators of STAT3 activity

PLOS ONE

Dear Dr. Wennerberg,

Thank you for submitting your manuscript to PLOS ONE. After careful consideration, we feel that it has merit but does not fully meet PLOS ONE’s publication criteria as it currently stands. Therefore, we invite you to submit a revised version of the manuscript that addresses the points raised during the review process.

Remarks and critiques advanced by three Reviewers are appended to this letter. 

In essence, there is a need for a better understanding of the quality and significance of the results shown. This can be achieved by simply amending the text with requested clarifications, additional comments, inclusion of new references and, should it be the case, a convincing rebuttal. Response to point 1 (Reviewer 1) can be omitted as the Discussion paragraph (line 348 to line 357) can be considered satisfactory. With regard to point 2 (Reviewer 1), please consider combining Fig. S5 with Fig.5 (and properly rearrange the text) should that be a more effective option to present available results.  

We would appreciate receiving your revised manuscript by Feb 16 2020 11:59PM. To enhance the reproducibility of your results, we recommend that if applicable you deposit your laboratory protocols in protocols.io, where a protocol can be assigned its own identifier (DOI) such that it can be cited independently in the future. For instructions see: http://journals.plos.org/plosone/s/submission-guidelines#loc-laboratory-protocols

We look forward to receiving your revised manuscript.

Kind regards,

Alessandro Datti, Ph.D.

Academic Editor

PLOS ONE

Journal Requirements:

Reviewers' comments:

Reviewer's Responses to Questions

**Comments to the Author**

1. Is the manuscript technically sound, and do the data support the conclusions?

Reviewer #1: Yes

Reviewer #2: Yes

Reviewer #3: Yes

2. Has the statistical analysis been performed appropriately and rigorously? 

Reviewer #1: Yes

Reviewer #2: I Don't Know

Reviewer #3: Yes

3. Have the authors made all data underlying the findings in their manuscript fully available?

Reviewer #1: Yes

Reviewer #2: Yes

Reviewer #3: Yes

4. Is the manuscript presented in an intelligible fashion and written in standard English?

Reviewer #1: Yes

Reviewer #2: Yes

Reviewer #3: Yes

5. Review Comments to the Author

Reviewer #1: The study authored by Parri et al reports on the identification of partially novel STAT3-regulators. The manuscript is very well written, the figures are clearly represented and overall the study is easy to read and to follow. Especially the description of the materials and methods section is very detailed. The manuscript shows a highly interesting and comprehensive data-set using a sophisticated screening assay (measuring viability, cytotoxicity and STAT3 transcriptional activity) targeting more than 1000 genes and narrowing them down to 7 targets regulating STAT3 activity.

There are still a few open questions:

1. Celltiter-Glo and CellTox Green measure viability and cytotoxicity, respectively. It is thus anticipated and was recently nicely shown that these two assays should complement each other and basically show opposite result (please see https://www.ncbi.nlm.nih.gov/books/NBK540958/). It would be very interesting to know, what the authors think that the reasons might be for the observed differences in cytotoxicity, while the viability stays unchanged when treating the cells with different chemicals and inhibitors (Figure 1).

2. As CDK8, CDC7 and CSNK2A1 represent serine/threonine kinases, it is puzzling why the authors chose to analyse the effect of the respective inhibitors on STAT3-Y705 phosphorylation, rather than STAT3-S727 phosphorylation. It would be advisable to add the effect of kinase inhibition on STAT3-S727 phosphorylation (according to Figure 5). And to make the story complete, please add the inhibitor treatment and STAT3-pY/pS-analysis on IL-6-treated STAT3-WT cells.

In line 335, the authors mention that CDK8-i causes the strongest reduction of pY705 posphorylation, but Figure 5 actually shows that CDC7-i is equally potent.

It is intriguing that serine kinase inhibition impacts on tyrosine-phosphorylation. It would be highly interesting to know the authors‘ thoughts about the reason for the reduced Y705-STAT3 phosphorylation upon serine-kinase inhibition.

3. With regards to the observation, that some of the inhibitors showed a clear kinetic in the STAT3 reporter activity assays in STAT3-mutant cells, the authors conclude that this suggests a difference in protein phosphorylation turnover. Could this observation also be a result of constitutive STAT3 phosphorylation in the mutant cell line compared to de-novo-STAT3 activation upon IL-6 in WT cells? Or could this point towards an indirect effect of kinase-inhibition on STAT3 activity? It will be interesting to hear the author’s view on this.

4. The reporter assays shown in Figure 4 suggest that in most cases, kinase inhibition has a more potent effect in STAT3-Y740F cells compared to IL-6-treated STAT3-WT cells. Please discuss these findings in the manuscript. These observations are particularly interesting, as Figure 1 shows that constitutively activated STAT3 protects cells from cell toxicity.

5. The authors mention that the data in Figure 3 shows that only PI4KII inhibited STAT3 and STAT1 activity. To this reviewer, the effect of CDK8 KD looks very similar in this assay, and also CSK KD seems to have some (counterintuitive) effect on STAT1-activity. Please explain the rational why the authors do not mention these, and please add and/or discuss. Additionally, the authors claim that the KD generally caused stronger inhibition of WT STAT3 activity than mutant STAT3. It is not clear how the authors came to this conclusion. Please add the respective data to support this conclusion.

6. In line, Figure 4 shows that the inhibition of CSNK2 and DDR2 have greater effects on STAT3-reporter activity in STAT3-Y640F mutant cells compared to STAT3-WT cells. Please discuss the discrepancy to the previous point.

7. In lines 286-289 the authors mention that they see „stronger reduction“ or reporter activity in STAT3-Y740F cells than in STAT3-WT cells, but the major difference is not the reporter activity, but the kinetics of it. The authors may consider changing their wording.

8. In the discussion in line 387 the authors mention that CDK8 has not been linked to STAT3 before, although afterwards they clearly discuss the previous findings by Bancerek et al. Likewise, also CSK has been previously (indirectly) linked to STAT3 phosphorylation.

Minor points:

1. Please explain the molecular details and functional consequences of the STAT3(Y640F) mutation in the manuscript (especially with regards to Y705, S727 phosphorylation and target gene transcription).

2. Please indicate whether the mycoplasma-tests were positive/negative.

3. Please show the data mentioned in line 215 and 219 (reporter assay signal in STAT3-Y740F versus IL-6-stimulated STAT3-WT cells; STAT3 siRNA silencing).

4. Is there a reason, why the authors do not mention PI4KII and PTPRH in the abstract?

5. The sentence in lines 273-275 is difficult to understand.

6. For the purpose of consistency, please add the name of the CDC7 inhibitor in line 289.

7. Please indicate in Figure 5, which samples have been analysed after 48 or 72 hours, respectively.

8. In line 384 the authors mention that the six kinases and one phosphatase regulate EITHER STAT3-mutant OR STAT3-WT in HEK cells, but it should read AND/OR.

9. In the legend of Figure S2: Please add „STAT3“- reporter activity... (line 618).

10. In Figure 1C and 1C the axis of the GSK-461364 inhibior are not set to 100.

11. The data in Figure 2D is displayed in a rather complicated manner. Maybe the authors find an easier way to plot their data.

Reviewer #2: The supplemental files contain processed data, but should be labeled as to their content, and how data was normalized or processed. The supplemental files need legends and supplemental text. I did not see it?

Reviewer #3: STAT3 mediates signalling downstream of cytokine and growth factor receptors where it acts as a transcription factor for its target genes, including oncogenes and cell survival regulating genes. STAT3 has been found to be persistently activated in many types of cancers, primarily through its tyrosine phosphorylation (Y705), and it is often taken as an target gene for the cancer therapeutic agents. In this paper, the authors find that constitutive STAT3 activation protects cells from cytotoxic drug responses of several drug classes. After performing a kinase and phosphatase siRNA screen with cells expressing either a hyperactive STAT3 mutant or IL6-induced wild type STAT3, the authors suggest that inhibiting the kinases (CDC7, CSNK2, DDR2 and CDK8 may provide strategies for dampening STAT3 activity in cancers. The manuscript is well organized, but I would like to encourage the authors to do some improvements.

1. In Figure 3, what is the meaning of the three bar columns in each group? And why are there only two bars in the CDK8 group of Fig 3C?

2. It is very interesting that 1μM ruxolitinib could not markedly reduce pY705 STAT3 level in STAT3(Y640F) expressing cells after 48 or 72 hours, but had a strong inhibitory effect on STAT3(wt) Y705 phosphorylation already after 4 hours. Therefore, I suggest that the authors use figure 4S as the formal figure and discuss this phenomenon further.

3. If the authors could draw a graph of these small molecular inhibitors on the targets of STAT3 signalling, it would be more helpful for the readers to understand the paper.

6. PLOS authors have the option to publish the peer review history of their article (what does this mean?). If published, this will include your full peer review and any attached files.

Reviewer #1: No

Reviewer #2: No

Reviewer #3: No

---

## [Author Response · Author response to Decision Letter 0]

4 Mar 2020

Reviewer #2? (comments sent as separate MS Word document without reference to Reviewer number)

In these experiments, the authors have searched for previously unexplored druggable targets that inhibit STAT3 signalling, in cells with both WT and mutant STAT3. They show that constitutively active STAT3 protects cells from cytotoxic responses to several

drug classes. They use transcription reporters in cells expressing STAT3 WT and STAT3(Y640F) in an RNAi screen, and identified kinases that altered activity. They use small molecule inhibition of several of kinase hits, which also blocked STAT3 activity, and conclude that targeting these kinases may therefore be an alternative strategy to inhibit STAT3 in cancers and other pathological conditions.

Comments:

The Introduction is informative with a nice rational, and its ends with a brief description of type of experiments. Previous work on drug sensitivity of STAT3 is referenced well. The experimental results are described in detail and look reasonable, but need a better take-away and what is the further work needed. Its is not clear what is novel in this study. In the discussion, they note that “in the siRNA screen three kinases and one phosphatase were identified (CDC7,415 PI4KII, DDR2 and PTPRH), which have not previously been linked directly to STAT3”. A number of other hits were discussed as previously identified regulators of STAT3, which adds confidence to their iRNA screen. 

We feel that the novelty of our study lies in the comprehensive profiling of kinases and phosphatases that may regulate STAT3 activity. In this process (as mentioned by the reviewer), we identify both previously identified STAT3-regulalting proteins and ones that have not been linked to STAT3 activity. Importantly, we show that small molecule inhibitors of several of these targets can also induce STAT3 inhibition, opening up for follow-up studies in more complex models. To further emphasize these conclusions, we have adjusted the closing statement in the discussion (lines 427-435).

However, drug screening does not appear to yield specificity for inhibition of activated STAT3(Y640F) compared to IL6 stimulated WT, nor for viability of STAT3(Y640F) cancer vs normal cells. The manuscript should be edited for clarity in this regard.

We agree with the reviewer that siRNA and small molecule inhibition in general did not look significantly different between STAT3(wt) and STAT3(Y640F) expressing cells for the tested hit genes. This means that other than CSK, we did not identify clear STAT3 mutant-selective regulators. On the other hand, it means that the identified targets can be explored both in cancers with STAT3 mutations and in cancers with hyperactive wild-type STAT3. We have clarified this in the text at lines 226-232 and 375-379.

Other comments:

1. The Discussion is long and the 1st paragraph repeats previous ideas from the intro. 

The discussion has been shortened and we have tried to avoid repetition of ideas from the introduction.

2. There too much description of the result in the Figure legends, which are in the Results section. Figure Legends should be concise, noting the method, and any specifics; the number biological and technical repeats and statistical significance and method. 

We have edited the figure legends accordingly.

3. How many independent experiments and what do error bars represent?

We have included this information to the figure legends.

4. The Figure legends should all be together after the references in manuscripts. 

Figure legends have been moved to after the references. 

5. Fig 2B “at least” is 2 words. 

This has been corrected.

6. Fig 2A and B could be simply described in the methods, as they are repetitive and contain a lot of text.

We have removed Fig 2A and 2B from the main figures.

Reviewer #1: The study authored by Parri et al reports on the identification of partially novel STAT3-regulators. The manuscript is very well written, the figures are clearly represented and overall the study is easy to read and to follow. Especially the description of the materials and methods section is very detailed. The manuscript shows a highly interesting and comprehensive data-set using a sophisticated screening assay (measuring viability, cytotoxicity and STAT3 transcriptional activity) targeting more than 1000 genes and narrowing them down to 7 targets regulating STAT3 activity.

There are still a few open questions:

1. Celltiter-Glo and CellTox Green measure viability and cytotoxicity, respectively. It is thus anticipated and was recently nicely shown that these two assays should complement each other and basically show opposite result (please see https://www.ncbi.nlm.nih.gov/books/NBK540958/). It would be very interesting to know, what the authors think that the reasons might be for the observed differences in cytotoxicity, while the viability stays unchanged when treating the cells with different chemicals and inhibitors (Figure 1).

We have now addressed this point in the discussion (lines 330-333). We have previously described that in a strong viability response (with relatively fast-growing cells), it is often difficult to separate cytotoxic from cytostatic responses [1]. However, the viability inhibition curves in general in Fig 1 go a bit deeper in the case where greater cytotoxicity is induced. 

2. As CDK8, CDC7 and CSNK2A1 represent serine/threonine kinases, it is puzzling why the authors chose to analyse the effect of the respective inhibitors on STAT3-Y705 phosphorylation, rather than STAT3-S727 phosphorylation. It would be advisable to add the effect of kinase inhibition on STAT3-S727 phosphorylation (according to Figure 5). And to make the story complete, please add the inhibitor treatment and STAT3-pY/pS-analysis on IL-6-treated STAT3-WT cells.

This is an important point and we have therefore added data also on pS727 levels with all inhibitor treatments (Figure 5).

In line 335, the authors mention that CDK8-i causes the strongest reduction of pY705 phosphorylation, but Figure 5 actually shows that CDC7-i is equally potent.

It is intriguing that serine kinase inhibition impacts on tyrosine-phosphorylation. It would be highly interesting to know the authors‘ thoughts about the reason for the reduced Y705-STAT3 phosphorylation upon serine-kinase inhibition.

We agree that CDC7 inhibition is similar in its potency to reduce pY705 levels to CDK8 inhibition and we have therefore have corrected this on lines 316-317. 

Our phospho-STAT3 western blotting data suggest that all hit genes likely are indirect regulators of STAT3 (not directly phosphorylating or dephosphorylating STAT3) and therefore assume that the signals that are affected by CDC7 or CDK8 inhibition ultimately lead to reduced phosphotyrosine levels through altered regulation of tyrosine kinases/phosphatases. We have addressed this in in the results and discussion (lines 316-322 and lines 372-380).

3. With regards to the observation, that some of the inhibitors showed a clear kinetic in the STAT3 reporter activity assays in STAT3-mutant cells, the authors conclude that this suggests a difference in protein phosphorylation turnover. Could this observation also be a result of constitutive STAT3 phosphorylation in the mutant cell line compared to de-novo-STAT3 activation upon IL-6 in WT cells? Or could this point towards an indirect effect of kinase-inhibition on STAT3 activity? It will be interesting to hear the author’s view on this.

We appreciate this point and agree that it is also possible that the difference between STAT3(wt) and STAT3(Y640F) expressing cells is due to the difference in assays. We have therefore addressed this in the manuscript on lines 254-264.

4. The reporter assays shown in Figure 4 suggest that in most cases, kinase inhibition has a more potent effect in STAT3-Y740F cells compared to IL-6-treated STAT3-WT cells. Please discuss these findings in the manuscript. These observations are particularly interesting, as Figure 1 shows that constitutively activated STAT3 protects cells from cell toxicity.

After reviewing the data, we find that only the DDR2 inhibitor regorafenib has a more potent effect in STAT3(Y640F) cells at later time points. Other treatments show highly similar effects in STAT3 mutant and wt cells (in the attached figure blue and green lines at 72h). At shorter time points (gray and red lines at 4h) the compounds are more effective in IL6-induced STAT3(wt)-cells. The figure below has data from manuscript Fig 4 and Fig S2 and clarifies this. We have added this to the manuscript to lines 254-263 and 279-280.

In figure 4 we observed an error in the silmitasertib viability curve that has been corrected. Furthermore, we added data from one more biological replicate to the figure and edited the figure legend accordingly.

5. The authors mention that the data in Figure 3 shows that only PI4KII inhibited STAT3 and STAT1 activity. To this reviewer, the effect of CDK8 KD looks very similar in this assay, and also CSK KD seems to have some (counterintuitive) effect on STAT1-activity. Please explain the rational why the authors do not mention these, and please add and/or discuss. 

We agree that the CDK8 KD is very similar in in STAT1 assay, but we could not conclude an effect by CDK8 KD on STAT1 transcriptional activity because of having tested only two siRNAs. We have clarified this in the text on lines 241-243. 

The counterintuitive CSK KD on effect to STAT3 vs STAT1 could be due to that SRC is activated by IFNγ and is involved in the activation of STAT3 but not STAT1 [2]. However, the effects we see on STAT1 transcriptional activity by CSK knockdown is not consistent enough across the tested siRNAs to draw a conclusion that it causes an inhibition rather than an activation. We have clarified this in the manuscript on lines 241-242 and in discussion lines 376-380.

Additionally, the authors claim that the KD generally caused stronger inhibition of WT STAT3 activity than mutant STAT3. It is not clear how the authors came to this conclusion. Please add the respective data to support this conclusion.

We observed stronger general inhibition of STAT3(wt) in the primary screen (Figure 2A and figure below), but this observation didn’t carry through in the validation screen and the same difference was not observed in the final hits. Therefore, we have removed the claim from the manuscript (from figure legend 3).

6. In line, Figure 4 shows that the inhibition of CSNK2 and DDR2 have greater effects on STAT3-reporter activity in STAT3-Y640F mutant cells compared to STAT3-WT cells. Please discuss the discrepancy to the previous point.

The previous comment was about the siRNA screen as whole as mentioned earlier, we realized that this effect was not consistent through the follow-up screens and we therefore removed that statement. As discussed in point 4, in figure 4, although it may first look like several of the small molecule inhibitors were effective at inhibiting STAT3 reporter activity in STAT3 mutant cell than in STAT3 wt cells, only the DDR2 inhibitor has a clear selective effect. 

7. In lines 286-289 the authors mention that they see „stronger reduction“ or reporter activity in STAT3-Y740F cells than in STAT3-WT cells, but the major difference is not the reporter activity, but the kinetics of it. The authors may consider changing their wording.

It is true that the major difference is the kinetics of the two cell lines. We have changed the wording in the manuscript (lines 269-271).

8. In the discussion in line 387 the authors mention that CDK8 has not been linked to STAT3 before, although afterwards they clearly discuss the previous findings by Bancerek et al. Likewise, also CSK has been previously (indirectly) linked to STAT3 phosphorylation.

We agree that these were previous links that we had failed to report and the text has now been corrected (lines 374-375).

Minor points:

1. Please explain the molecular details and functional consequences of the STAT3(Y640F) mutation in the manuscript (especially with regards to Y705, S727 phosphorylation and target gene transcription).

The mutations in the SH2 domain, STAT3(Y640F) result in increased hydrophobicity of the STAT3 dimerization site (pY+3 pocket) that results in a more stable form of STAT3 dimer, constitutive Y705 phosphorylation, enhanced nuclear stability and increased transcriptional activity. The STAT3(Y640F) mutation does not have an effect on S727 phosphorylation [3]. We have clarified this in the manuscript (lines 45-49). 

2. Please indicate whether the mycoplasma-tests were positive/negative.

Mycoplasma tests were negative and we have clarified this in the manuscript (lines 87-88)

3. Please show the data mentioned in line 215 and 219 (reporter assay signal in STAT3-Y740F versus IL-6-stimulated STAT3-WT cells; STAT3 siRNA silencing).

We have made supplementary figure using raw luciferase data addressing the data mentioned in lines 215 – 219 (Fig S1) and the text is edited accordingly. Furthermore, we have labelled separately siSTAT3 and siDEATH in figure 2A where the positive control is siSTAT3 and assay control is siDEATH.

4. Is there a reason, why the authors do not mention PI4KII and PTPRH in the abstract?

The reason why we initially did not mention PI4KII and PTPRH in the abstract was that we did not have selective small molecule inhibitors against these proteins. We have included them now in the abstract.

5. The sentence in lines 273-275 is difficult to understand.

The sentence: “Generally, the reporter activity inhibition occurred more slowly in STAT3(Y640F) expressing cells than the STAT3(wt) cells, suggesting that the STAT3(Y640F) protein phosphorylation turnover is slower.” Has been edited to: “We observed that the reporter activity inhibition occurred more slowly in STAT3(Y640F) expressing cells than the STAT3(wt) cells, suggesting that the STAT3(Y640F) protein de-phosphorylation is slower.“ Lines 254-260.

6. For the purpose of consistency, please add the name of the CDC7 inhibitor in line 289.

We have corrected this to the text.

7. Please indicate in Figure 5, which samples have been analysed after 48 or 72 hours, respectively.

We have labelled the samples accordingly in figure 5. 

8. In line 384 the authors mention that the six kinases and one phosphatase regulate EITHER STAT3-mutant OR STAT3-WT in HEK cells, but it should read AND/OR.

The text has been edited according to the suggestion.

9. In the legend of Figure S2: Please add „STAT3“- reporter activity... (line 618).

We have edited the Fig S2 legend according to the suggestion.

10. In Figure 1C and 1C the axis of the GSK-461364 inhibior are not set to 100.

The GSK-461364 data had points outside the range of the ranges of the other data sets and we had therefore set a different scale on the y-axis for this inhibitor. As the reviewer suggests, it makes sense to have the graphs coherent and we have therefore now set the y-axis maximum to 125 for all the inhibitors in figure 1C and 1D.

11. The data in Figure 2D is displayed in a rather complicated manner. Maybe the authors find an easier way to plot their data.

Figure 2D (now Figure 2B) has been simplified.

Reviewer #2: The supplemental files contain processed data, but should be labeled as to their content, and how data was normalized or processed. The supplemental files need legends and supplemental text. I did not see it?

We have added supplemental legends and text explaining the data in supplemental files and corrected the labelling of the files. 

Reviewer #3: STAT3 mediates signalling downstream of cytokine and growth factor receptors where it acts as a transcription factor for its target genes, including oncogenes and cell survival regulating genes. STAT3 has been found to be persistently activated in many types of cancers, primarily through its tyrosine phosphorylation (Y705), and it is often taken as an target gene for the cancer therapeutic agents. In this paper, the authors find that constitutive STAT3 activation protects cells from cytotoxic drug responses of several drug classes. After performing a kinase and phosphatase siRNA screen with cells expressing either a hyperactive STAT3 mutant or IL6-induced wild type STAT3, the authors suggest that inhibiting the kinases (CDC7, CSNK2, DDR2 and CDK8 may provide strategies for dampening STAT3 activity in cancers. The manuscript is well organized, but I would like to encourage the authors to do some improvements.

1. In Figure 3, what is the meaning of the three bar columns in each group? And why are there only two bars in the CDK8 group of Fig 3C?

Each bar represents different siRNAs targeting the labeled gene (i.e Hs_CDC7_1, Hs_CDC7_2, Hs_CDC7_5). The used siRNAs are listed in supporting file S3. The bars are matched so that in the figures 3 A, B, C individual siRNAs are in same order, with exception with siCDK8 in Fig 3C where only the first two siRNAs are matched. The figure has been edited and the different siRNAs are labelled to clarify this. 

There are only two bars for CDK8 because we only had 2 siRNAs for this gene. 

2. It is very interesting that 1μM ruxolitinib could not markedly reduce pY705 STAT3 level in STAT3(Y640F) expressing cells after 48 or 72 hours, but had a strong inhibitory effect on STAT3(wt) Y705 phosphorylation already after 4 hours. Therefore, I suggest that the authors use figure 4S as the formal figure and discuss this phenomenon further.

The difference between the effects on the wild type vs. mutant STAT3 likely comes from that we are comparing de novo vs. persistent phosphorylation. In the STAT3(wt) expressing cells STAT3 is activated by the addition of IL6 after the ruxolitinib addition whereas in the STAT3(Y640F) the Y640F is already phosphorylated before adding ruxolitinib. Therefore, the effects on the wt phosphorylation are assessing the impact of directly blocking JAK-dependent phosphorylation of STAT3 (caused by IL6 stimulation). In the mutant case, we assess the impact on turnover of the phosphorylation (dephosphorylation followed by re-phosphorylation) of STAT3. Further enhancing the differential effect, The STAT3(Y640F) mutant has been described to cause a more stable dimer than the wild type protein, causing a slower rate of dephosphorylation. Since the pY705 activation by IL6 in STAT3(wt) cells is fairly fast and short-lived we used a 20-minute IL6 stimulation prior to lysing the cells for western blots and a 3-hour stimulation for the luciferase reporter readout. Ruxolitinib serves in this study as control for de novo vs. constitutive STAT3 activation. 

To clarify these differences between the wt and mutant-expressing cells, we have addressed this in the text on lines 45-49 and 254-263. 

3. If the authors could draw a graph of these small molecular inhibitors on the targets of STAT3 signalling, it would be more helpful for the readers to understand the paper.

We have added a summary figure 6 of the siRNA screen hits and inhibitors targeting them.

References

1. Gautam P, Karhinen L, Szwajda A, Jha SK, Yadav B, Aittokallio T, et al. Identification of selective cytotoxic and synthetic lethal drug responses in triple negative breast cancer cells. Molecular cancer. 2016;15(1):34. Epub 2016/05/12. doi: 10.1186/s12943-016-0517-3. PubMed PMID: 27165605; PubMed Central PMCID: PMCPMC4862054.

2. Qing Y, Stark GR. Alternative activation of STAT1 and STAT3 in response to interferon-gamma. The Journal of biological chemistry. 2004;279(40):41679-85. Epub 2004/07/31. doi: 10.1074/jbc.M406413200. PubMed PMID: 15284232.

3. Pilati C, Amessou M, Bihl MP, Balabaud C, Nhieu JT, Paradis V, et al. Somatic mutations activating STAT3 in human inflammatory hepatocellular adenomas. The Journal of experimental medicine. 2011;208(7):1359-66. doi: 10.1084/jem.20110283. PubMed PMID: 21690253; PubMed Central PMCID: PMC3135371.

---

## [Decision Letter · Decision Letter 1]

10 Mar 2020

Identification of novel regulators of STAT3 activity

PONE-D-19-33342R1

Dear Dr. Wennerberg,

We are pleased to inform you that your manuscript has been judged scientifically suitable for publication and will be formally accepted for publication once it complies with all outstanding technical requirements.

With kind regards,

Alessandro Datti, Ph.D.

Academic Editor

PLOS ONE

Additional Editor Comments (optional):

Reviewers' comments:

Reviewer's Responses to Questions

**Comments to the Author**

1. If the authors have adequately addressed your comments raised in a previous round of review and you feel that this manuscript is now acceptable for publication, you may indicate that here to bypass the “Comments to the Author” section, enter your conflict of interest statement in the “Confidential to Editor” section, and submit your "Accept" recommendation.

Reviewer #1: All comments have been addressed

2. Is the manuscript technically sound, and do the data support the conclusions?

Reviewer #1: Yes

3. Has the statistical analysis been performed appropriately and rigorously? 

Reviewer #1: Yes

4. Have the authors made all data underlying the findings in their manuscript fully available?

Reviewer #1: Yes

5. Is the manuscript presented in an intelligible fashion and written in standard English?

Reviewer #1: Yes

6. Review Comments to the Author

Reviewer #1: (No Response)

7. PLOS authors have the option to publish the peer review history of their article (what does this mean?). If published, this will include your full peer review and any attached files.

Reviewer #1: No

---

## [Editor Report · Acceptance letter]

12 Mar 2020

PONE-D-19-33342R1 

Identification of novel regulators of STAT3 activity 

Dear Dr. Wennerberg:

I am pleased to inform you that your manuscript has been deemed suitable for publication in PLOS ONE. Congratulations! Your manuscript is now with our production department. 

With kind regards,

on behalf of

Dr. Alessandro Datti 

Academic Editor

PLOS ONE